# Cognitive Modeling for Human-Robot Value Soft Alignment

## Abstract

In the domain of human-robot symbiosis, it is of utmost importance for robots to display intelligent behavior. This encompasses the capability to deduce implicit information to predict the true intentions of humans. Traditionally, humans are perceived as flawless, with their decisions acting as the standards that robots should strive to align with. However, this raises a pertinent question: What if humans make mistakes? In this research, we present a unique task, termed "value soft alignment". This task involves the proactive identification of potentially hazardous or unsuitable human actions before the manifestation of their repercussions, and the provision of situationally appropriate suggestions. To facilitate this task, we have constructed a dataset for model training and testing, comprising two types of data: simulated human behaviors and collected human behaviors. We propose a value-driven cognition model to represent the understanding of human behavior, followed by a two-stage method that consists of 1) the prediction of value-based long-term intention and 2) the comparison between the long-term intention and the short-term immediate action intention. Experimental results indicate that the value-driven cognition model can assist robots in comprehending human behavioral patterns over both long-term and short-term durations, and thus the robot can offer sensible recommendations for a majority of scenarios based on the consistency between long-term and short-term intentions of humans.

## 1 Introduction

The symbiotic relationship (Sandini et al., 2018) between humans and robots represents the fundamental paradigm for the future coexistence of artificial intelligence (AI) agents and humans. This paradigm necessitates that robots demonstrate advanced cognitive abilities (Clark & Grush, 1999) and intelligent behaviors (Rahwan et al., 2019), thereby enabling them to address intricate issues prevalent in human society. Cognitive robots (Levesque & Lakemeyer, 2008), which have significantly benefited from the swift advancements in AI, are capable of providing assistance by deducing human intentions through multi-modal perception. This has emerged as a primary method for robots to serve human society. Consequently, the accurate comprehension of genuine human intentions and the execution of tasks that align with the maximum human value (Rokeach, 2008) are pivotal research problems in the field of cognitive robotics.

At present, a significant portion of research on intention understanding (Blakemore & Decety, 2001; Fogassi et al., 2005; Tomasello, 2023) concentrates on immediate intentions by analyzing motion sequences. This approach aids in accomplishing tasks in which humans may need help. It operates on the assumption that a human's immediate intention aligns with their values that determine the human behavioral patterns (Schwartz, 2012), without considering the potential for inappropriate actions or even mistakes. For example, an individual engrossed in complex cooking tasks might forget an appointment with a friend. In such a scenario, an intelligent robot should recognize that continuing the cooking tasks is not the optimal choice. Instead, the robot should suggest that the human halt their current activity and attend the appointment to maintain their reputation as a punctual person. If the robot merely assists with the cooking tasks based on its observations without suggesting the appointment, it could be perceived as lacking intelligence. Even advanced large language models, such as ChatGPT (Aljanabi et al., 2023) and GPT-4 (OpenAI, 2023), may only account for the immediate situation in human-robot interaction. These AI models, as well as cognitive robots driven by these models, often overlook the influence of high-level human values, resulting in suboptimal

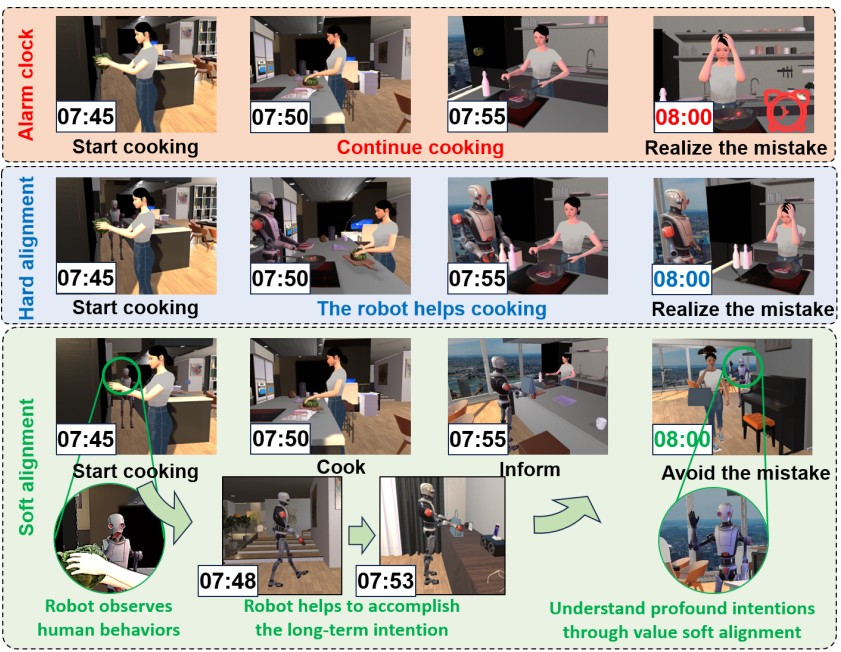

Figure 1: Comparison between hard value alignment and soft value alignment.

assistance or even no assistance when a human's immediate actions inadvertently violate long-term values.

To address the aforementioned issues, robots should be able to learn human values and corresponding behavioral patterns, as well as the immediate intentions reflected by short-term motion sequences. This would enable them to provide useful suggestions when individuals engage in actions that violate long-term intentions aligned with human values. We refer to this as "value soft alignment", as the robot's goal is to align with high-level values rather than every specific action intention. This introduces two challenges: firstly, constructing human value models via observation to represent human behavior patterns; secondly, determining whether the immediate action intention conflicts with value-driven long-term behavioral patterns. The first challenge necessitates a substantial amount of lifelog data for the machine learning model to extract human behavioral patterns. However, collecting this type of data is challenging, and there is no readily available dataset that supports this task. The second challenge involves predicting both the immediate action intention (short-term intention) and the value-driven human behavioral patterns (long-term intention), and assessing whether these two types of intentions conflict with each other, which increases the instability in overall judgment results. In summary, a cognitive structure suitable for representing complex human intentions should be constructed, by which the robot can provide meaningful assistance services.

In this paper, we construct a continuous, long-term human life recording dataset by combining data synthesis based on large language models with human data recording via virtual simulation. This dataset supports model training and testing, providing the fundamental data required to address the mentioned value soft alignment task. We propose to build a value-driven cognitive model for robots to understand human behaviors, which allows the robot's real-time detection of whether the anticipated outcomes of short-term human behavior align with the value-driven long-term behavioral patterns. This enables robots to provide reasonable suggestions as early as possible before the potential adverse consequences of human behavior emerge.

Our contributions are three-fold:

- We construct a dataset for value soft alignment tasks by combining large language model-based data synthesis with virtual simulation-based data recording.

- We build a value-driven cognitive model to tackle the proposed "value soft alignment" tasks, which allows the robot to learn long-term behavioral patterns guided by human values and provide immediate suggestions for potentially unreasonable short-term behaviors.

- Our experimental results verify that the proposed cognitive model can fit the value-based human behavioral patterns and help to indicate potential suboptimal behavior and even mistakes.

## 2 RELATED WORKS

### 2.1 DATASETS OF HUMAN ACTIONS

The study of human behavior often involves segmenting continuous motion sequences into discrete, atomic actions. This approach is exemplified by the AVA dataset (Gu et al., 2018). Concurrently, the interactions between humans and objects provide a wealth of information about human behavior. This has led to the creation of datasets like LEMMA (Jia et al., 2020), which offers action labels for human-object interactions, thereby facilitating the understanding of multi-agent, multi-task activities in everyday life. For robots, which typically perceive their environment from an ego-centric viewpoint, ego-centric tasks (Jia et al., 2022) are crucial for understanding human behavior.

### 2.2 PREDICTING ACTION INTENTIONS

The prediction of human intentions has long been a focal point in the field of cognitive robotics. Puig et al. introduced a simulator known as VirtualHome-Social, which incorporates a challenge related to intention prediction (Puig et al., 2021). The authors devised five tasks, each representing a different type of household activity. These tasks primarily concentrate on the spatial relationships between objects, without taking into account the specific fluent representations of the objects.

Human actions can be bifurcated into categories such as intentional and unintentional actions. Epstein et al. posited that the velocity of an action is intrinsically linked to its intentionality (Epstein et al., 2020). Actions executed at a slower pace are likely to be intentional, whereas actions performed swiftly are likely to be unintentional (Caruso et al., 2016). Accordingly, Epstein et al. introduced the OOPS dataset for the prediction of unintentional actions (Epstein et al., 2020). This dataset comprises 20,338 videos and is accompanied by a self-supervised algorithm that learns representations of intention. Given the intricate intentions inherent in human actions, a robot necessitates a cognitive structure (Kotseruba et al., 2016) to decipher complex human behaviors.

### 2.3 AUTOMATED TASK PLANNING

Various methodologies have been proposed to articulate the planning problem. The Planning Domain Definition Language (PDDL) is a widely adopted approach and has been the official language of the International Planning Competition (IPC) since 1998 (Aeronautiques et al., 1998). Over the past decades, researchers have sought to broaden this definition. In addition to PDDL, other languages such as Probabilistic PDDL (PPDDL) (Younes et al., 2005), Relational Dynamic Influence Diagram Language (RDDL) (Sanner et al., 2010), and Behavior Domain Definition Language (BDDL) (Li et al., 2021) have been proposed. For relatively complicated situations, hierarchical planners can be employed. For instance, an agent can plan how to assist its partner with partial observation (Puig et al., 2021). The planning process is divided into two stages: high-level planning and low-level planning. The high-level planner determines the optimal sub-goal, while the low-level planner takes this sub-goal as input and generates an executable action. Inspired by these works, the proposed "value soft alignment" problem may also be treated as a multi-hierarchy cognition problem.

## 3 VALUE SOFT ALIGNMENT

### 3.1 PROBLEM DEFINITION

**Value alignment** is typically required in human-robot collaboration (Yuan et al., 2022), as it fosters trust between humans and robots. However, this becomes more complex in the context of everyday life scenarios. We denote the case where an intelligent robot strictly follows humans' immediate intentions as **value hard alignment**. Many efforts have been spent on such challenges to predict human attention, intention, and immediate collaboration (Puig et al., 2020; Fathi & Rehg, 2013; Nan

et al., 2020). However, humans' immediate choices and actions may be suboptimal or even counter-productive when viewed in light of long-term behavioral patterns driven by human values. Figure 1 shows an example where human's immediate action (cooking) is in conflict with her long-term behavioral patterns (going to work). An intelligent robot is expected to align with the values reflected by these long-term behavioral patterns, rather than directly assisting humans with immediate tasks that may contradict long-term human values. Such a robot demonstrates the ability to softly align with human values, which we denote as **value soft alignment**.

### 3.2 CHALLENGES AND DESIDERATA

The aforementioned "value soft alignment" task involves the concurrent prediction of long-term and short-term intentions, as well as the identification of potential conflicts between these intentions. To tackle this problem, one basic requirement is to parse human actions and environmental states (such as states of objects in the house). Since object detection and action recognition algorithms have been widely discussed, we will ignore this part in this paper to make this work more compact. We attribute the rest of this problem to the following three challenges.

**Long-term observation.** We argue that humans' long-term behavior patterns are crucial factors in inferring their values, while such patterns are different from person to person. As mentioned in the related works, most datasets focus on short-term action data in several minutes or hours. They care about learning universal patterns among a community rather than individual patterns. Therefore, they may have a lot of data from different people but only a few records per person. However, to understand long-term behavior patterns, it is necessary to have long-term observations of humans individually, typically more than a fortnight. Thus, we list long-term observation as the first challenge. The observation should track each person as well as the surrounding environment for a relatively long time. The expected observation data set $\mathcal{O}$ is denoted in Equation 3.

$$\mathcal{O}_p = \{\mathcal{A}, \mathcal{S}, \mathcal{L} \,|\, t \in [t_0, t_n], p \in \mathcal{P}\} \tag{1}$$

where $\mathcal{A}$ denotes the action set observed from $t_0$ to $t_n$, $\mathcal{S}$ denotes the set of environment states observed from $t_0$ to $t_n$, $\mathcal{L}$ denotes the label set of each action-state data pair, $p$ denotes the index of the observed participant, and $\mathcal{P}$ denotes the participant set. Note that the label set could be the intention of each action annotated by humans or other useful information.

**Human value modeling.** Numerous facets intertwine to define the complex concept of human value, encompassing biological imperatives, psychological yearnings, ingrained behavioral patterns, and intricate personality traits, etc. Some facets take a long time to be observed. Modeling human value in this task imposes a significant expectation upon the agent: the ability to achieve alignment with human values over a long temporal horizon, transcending mere transient insights. The agent learns a function $\mathcal{F}_p : \mathcal{O}_p \rightarrow \mathcal{V}_p$ based on the observed behaviors, where $\mathcal{V}_p$ denotes the hidden value of the observations. In the realm of decision-making, a well-established conundrum emerges, wherein the agent must navigate the terrain of short-term gains (e.g., greedy algorithm) versus long-term benefits (e.g., resisting the allure of high-calorie foods while adhering to a dietary regimen). This dilemma serves as a litmus test for the agent's capacity to transcend conventional paradigms of value alignment, moving beyond the realm of immediate gratification. Different from value hard alignment, which mainly focuses on short-term value, the agents should have the ability to model human's long-term value.

**Conflict prediction.** Inferencing the intentions of others stands as a pivotal capability within the repertoire of human beings. It empowers individuals to proactively save others from making mistakes by dispensing valuable advice in advance. In this challenge, the agent is tasked with replicating and responding to this ability. For elucidation, we represent the sequence of actions required to realize a particular intention as $\vec{a} = (\mathbf{a_1}, \mathbf{a_2}, ..., \mathbf{a_n})$. On one hand, the agent is required to predict the short-term intention before humans complete the actions in $\vec{a}$. On the other hand, the agent is required to predict a long-term intention based on the human value model. Prediction of the conflicts between short-term and long-term intentions could be denoted as Equation 2:

$$P_{conflict} = P(\mathcal{D}(I_{long,t_k}, I_{short,t_k}) > th | \vec{O_t}), t \in [t_0, t_k] \tag{2}$$

where function $\mathcal{D}(\cdot)$ calculates the difference between $I_{long,t_k}$ and $I_{short,t_k}$ and $th$ is a threshold for reporting conflicts. $\vec{O_t}$ denotes sequence of observation from $t_0$ to $t_k$.

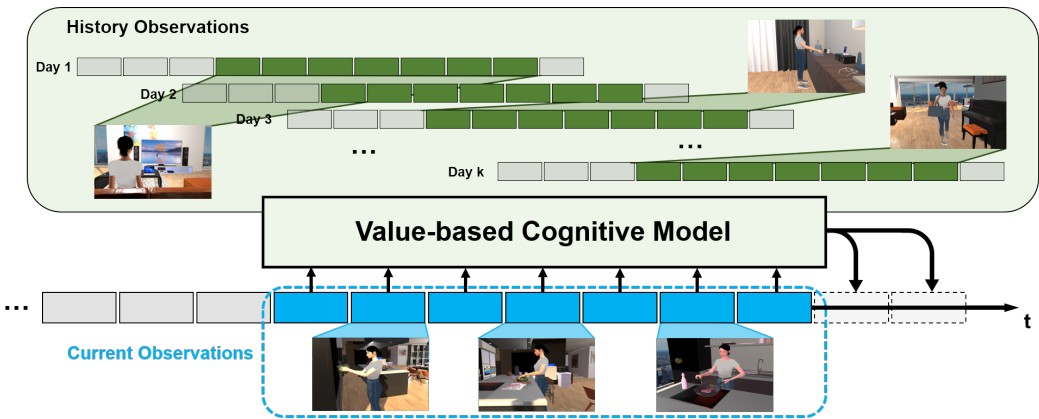

Figure 2: We design a system that models human value from long-term observation driven by a value-based cognitive model.

# 4 METHODS

## 4.1 DATA COLLECTION

To tackle the first challenge, we create two kinds of behavior collectors. On the one hand, behavior data directly collected from humans has advantages in terms of quality, but has disadvantages in the cost and collection difficulty. On the other hand, large language models (LLMs) are used for data generation and data annotation, which is cheap but has more uncertainty than human annotations. (Touvron et al., 2023; Wang et al., 2022; Chiang et al., 2023; Liu et al., 2023) We launch the two approaches for data collection and create two datasets: a simulated human behavior dataset and a collected human behavior dataset.

**Simulated human behaviors.** We prompt ChatGPT to generate simulated human behavior data. Within the prompt, we provide a detailed task description, constraints on actions, a list of interactable objects, descriptions of rooms within the house, as well as human and object states. The actions and initial states are specified to ensure the creation of a coherent and logical behavior plan. More specifically, we compel ChatGPT to adopt distinct personalities that encompass behavioral habits and preferences. We collected 19 descriptions from 19 participants through a questionnaire. These personalities serve as references for ChatGPT to generate behaviors. Additionally, we offer a few behavior examples that strictly adhere to a standardized format when generating behaviors. Furthermore, we define the initial states of all interactive objects. These states may be altered by future behaviors, including changes in object quantities and statuses. ChatGPT is tasked with simulating the role of a specific participant, generating a list of formatted descriptions of their daily behaviors within a specified date range.

Note that the list of objects, actions, and rooms given to ChatGPT is consistent with the one used in the collected human behavior dataset. They also share the same data format.

**Collected human behaviors.** In order to collect human data, we propose a data collection system built on Unity3D. This system features a virtual apartment comprising 6 rooms and a total of 206 interactable objects, including a living room, kitchen, bathroom, bedroom, study, and hall. Participants are asked to simulate their daily routines within this virtual apartment. They immerse themselves in the scene from a first-person perspective and can modify the states of the objects in the apartment through interactions. Participants are asked to select their actions using a user interface as if they were residing in their real-world homes. They specify the corresponding objects and label their high-level intentions. After initializing objects with predetermined initial states, identical to those provided to ChatGPT, the system diligently records a comprehensive set of behavior data. This includes the action taken, the associated intention, the action's start time, its duration, the involved object, the resulting state of that object after the action, and the participant's position within the apartment. All collected data is meticulously stored in JSON format. In anticipation of potential challenges such as insufficient memory or system downtime during the long-term data collection

sessions, we have also implemented a breakpoint resume function. This feature allows the system to restore the states right up to the point of the last shutdown.

## 4.2 Value-based Cognitive Model

Here we discuss the second challenge. Since value soft alignment is a cognitive process, we refer to cognitive architectures when modeling human value. There has been a variety of cognitive architectures such as Belief-Desire-Intention (BDI) (Harnad, 1990), Learning Intelligent Distribution Agent (LIDA) (Faghihi & Franklin, 2012), Simulation of the Mental Apparatus & Applications (SiMA) (Schaat et al., 2015), Adaptive Control of Thought-Rational (ACT-R) (Ritter et al., 2019), etc. AGI models demonstrate high-level cognitive intelligence when incorporating values (Peng et al., 2023). Consequently, a value-based cognitive model holds significant potential for managing complex tasks. Following these, we propose our value-based cognitive model.

Our modeling methodology addresses the task of value soft alignment through long-short term intention modeling. To comprehend the intricate intentions encapsulated in human behaviors, the model considers both **short-term** and **long-term intentions**. The model takes human behavior information and environmental states into account, including but not limited to the action name, its start time and start date, its duration, the related objects, state changes caused by the action, and human states. Observations of human behaviors are indexed by their start time. Using a feature-extract function $\mathcal{F}_{extract}(\cdot)$, the observations with multi-source heterogeneous data are converted into a high-dimensional feature domain. Each observation corresponds to a vector in the feature domain, denoted as $\vec{obs}$.

$$\vec{obs}_t = \mathcal{F}_{extract}(O_t), t \in [t_0, t_k] \tag{3}$$

The intention $\mathcal{I}_{t_k}$ starting from $t_k$ may relate to a continuous sequence of observations denoted as $\vec{O}|_{t_k}$. This model puts special attention on the time feature and consists of three layers of attention:

- **L1: action.** It models the transitions between $\vec{obs}$ to predict the next $\mathbf{a}_{t_{i+1}}$ along with the possible $\vec{obs}_{t_{i+1}}$ and the duration of $\mathbf{a}_{t_{i+1}}$. This layer serves as a foundation for L2.

- **L2: short-term intention.** The prediction of future actions along with the observed history actions formulates an action list $\vec{a} = (\mathbf{a_1}, \mathbf{a_2}, ..., \mathbf{a_n})$. This layer models the short-term intention $I_{short}$ behind $\vec{a}$, which is also known as immediate intention.

- **L3: long-term intention.** It predicts a list of possible intentions $\vec{I}_{long}$ based on observation over an extended time window, considering the daily routines and past intentions.

This model thereby facilitates the detection of conflicts between immediate actions and long-term values. With the aid of this cognitive model, agents can gain a profound understanding of human behaviors, representing a pivotal advancement in the pursuit of holistic human-AI interaction.

## 4.3 System Implementation

The overall structure of the system implementation is shown in Figure 3.

**Data pre-process.** The observation data is pre-processed before being fed into the neural networks. In the pre-processing module, observation data $O_t$ is serialized into a m-dimensional vector $\vec{S}_t \in \Re^m$. As GPT may generate ghost behavior data (examples are shown in the Appendix), we double-check the simulated data before serialization. Most illegal data is fixed in an automated manner and the rest is discarded. Actions, objects, and human positions are encoded into binary codes during the serialization. Start time, start date, and weekday number are emphasized as separate features. Action durations and object states are encoded as vectors and are concatenated with the aforementioned features. Intentions of the observation are encoded into a feature domain using Sentence-BERT (Reimers & Gurevych, 2019).

**Base networks.** Inspired by Wu et al. (2020), we build the value learning model with a simplified multi-head Transformer structure, which is suitable for time series forecasting tasks. The basic structure of the model is shown in Figure 3. We utilized a shallow transformer with only one encoder layer and one decoder layer. Since the amount of data is limited, deep neural networks can

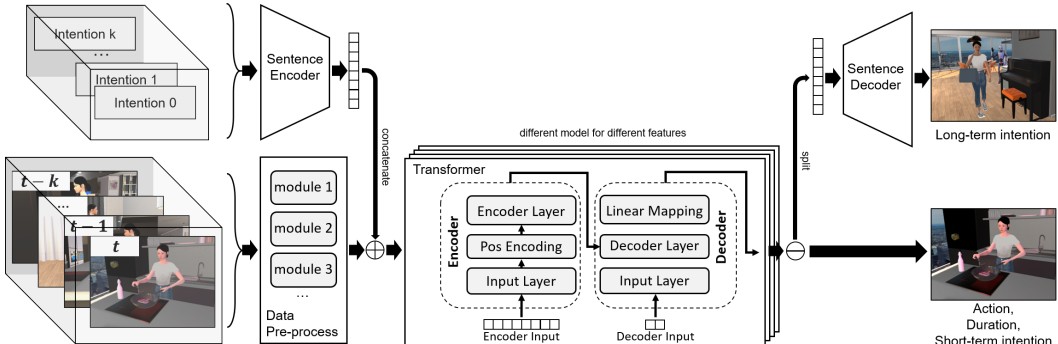

Figure 3: Algorithm structure. The temporal observation data is serialized before being fed into the neural networks. Intentions are singled out for feature encoding and decoding. The output of the system includes the prediction of actions, durations, short-term intentions, and long-term intentions.

easily overfit. Because we especially care about the timing of the observations, we utilized a 4-head transformer to make it more suitable for learning long-term temporal patterns.

According to our hierarchical cognitive model, we proposed four models that share the same structure. They learn the model of actions, action durations, short-term intentions, and long-term intentions individually. The only difference between them is the dimensions of the input layer and the decoder output.

We also proposed a conflict detection mechanism. We compare the similarity of the predicted long-term intention and the short-term intention. Note that we predict a list of potential long-term intentions in consideration of human uncertainties. If the short-term intention is not similar to any of the long-term intentions in the predicted list, the agent sends a query and proposes to help.

## 5 EXPERIMENTS

### 5.1 APPARATUS.

Four models are trained to learn actions, action durations, short-term intentions, and long-term intentions. For each model, we first train a basic model using the simulated human behavior dataset (based on 19 different personalities) generated by GPT-3.5 to learn the general value among people. This refers to the process of building a common sense model. Then, the basic model is fine-tuned to data sequences of the participants in the test set, one model per participant. It refers to the process of personalization. We finally test the performance of each model on the corresponding participant's data, as well as compare the performance across the baseline model, the basic model, and the fine-tuned models.

70% of the simulated dataset is used to train the basic model, while 10% for validation and 20% for testing. The test set contains data from 8 participants. Four of them are simulated using GPT-4 (P01-P04) while the other four are human participants (P05-P08). For each participant, we fine-tune the basic model using 70% of his/her data along with 10% used as validation. The remaining 20% data is used for testing.

### 5.2 TRAIN.

We train the four models respectively. The same input data formulation is utilized for predicting actions and durations: the action, its start time, its duration, the involved objects, the states of the objects after the action, and the participant's position within the apartment. The ground truth is the last observation of the input sequence. The input data for predicting short-term intentions is similar to that used to predict actions. The ground truth has one more label: the intentions related to the actions. The input data for predicting long-term intentions includes the labeled intentions.

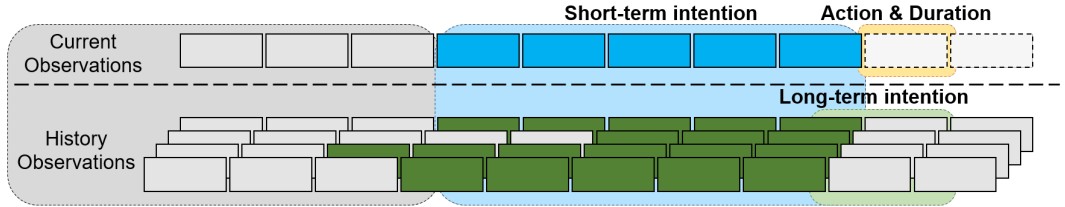

Figure 4: Four types of features to be modeled: (1) action, (2) action duration, (3) short-term intention, (4) long-term intention.

Models for predicting durations and the two kinds of intentions have the same structure. The model for actions is slightly different with two more linear layers after the decoder output, activated using the sigmoid function: $sigmoid(x) = \frac{1}{1+e^{-z}}$.

We utilized cross-entropy loss when training the action model. As for learning action durations, mean square error is utilized as loss. When learning the short and long-term intentions, a cosine embedding loss is utilized to compare the similarity between the predicted intention and the ground truth.

## 5.3   TEST.

We proposed a test set with two kinds of data: simulated human behaviors generated by GPT-4 (P01-P04) and collected human behaviors from real humans (P05-P08).

The baseline model has a similar basic structure but is trained end-to-end, thus, it doesn't follow the cognitive model we proposed. As value soft alignment is a novel task, we choose this baseline to evaluate the effectiveness of the value-based cognitive model we proposed.

The relative error between the predicted duration and the ground truth is utilized to evaluate the performance of duration prediction:

$$Error(pred, gt) = \frac{(pred - gt)^2}{gt^2 + \epsilon} \tag{4}$$

where $pred$ represents the predicted duration and $gt$ represents the ground truth. $\epsilon$ is a small compensation value in case the ground truth is zero. Since the duration value ranges from 0.5 to around a thousand, the difference between the prediction and the ground truth might be vastly influenced by the ground truth value (a 10-minute error per 900 minutes doesn't count for much but a 10-minute error per 1 minute is tremendous). We utilize Equation 4 to reduce such influences. Actions are seen as discrete symbols from a closed set $\mathcal{A}$ and are evaluated using top-1, top-3, and top-5 accuracies. The ground truth of intentions is described in natural language. To make the result clearer, we build an intention set $\mathcal{I}_{P_i}$ for each participant that contains all existing intentions of $P_i$, and compare the similarity of the predicted intention with intentions in $\mathcal{I}_{P_i}$. If the most similar one is the ground truth, this prediction is counted for top-1 accuracy. The top-3 and top-5 accuracies are calculated similarly.

## 5.4   RESULTS.

We compare the performance of the end-to-end model and our cognitive model among the three kinds of test data. The result is shown in Table 1. Overall, our method based on the hierarchical cognitive model has better performance than the end-to-end method. Different participants have their unique behavior patterns. The table also shows that the learning difficulty is different from person to person. P03 and P06 are considered difficult cases according to the result.

**Duration.** The results of the collected data are better than that of the GPT-4 data. We infer that the collected data has a smoother distribution than simulated data. **Action.** There are 26 kinds of actions in the dataset. The accuracies of simulated data are higher than that of the collected data. This infers that the simulated data has better consistency while the collected data from real humans has more randomness. **Short-term intention and long-term intention.** The performance of predicting long-term intention is slightly higher than short-term intention among a majority of participants. For most

Table 1: Model performance on data from different participants.

| | | duration | action | | | short_target | | | long_target | | |
|---|---|---|---|---|---|---|---|---|---|---|---|
| | | relative error | top-1 | top-3 | top-5 | top-1 | top-3 | top-5 | top-1 | top-3 | top-5 |
| End to end | P01 | 7.7658 | 0.14 | 0.56 | 0.81 | 0.09 | 0.30 | 0.40 | - | - | - |
| | P02 | 9.2312 | 0.16 | 0.37 | 0.45 | 0.13 | 0.21 | 0.27 | - | - | - |
| | P03 | 3.9362 | 0.17 | 0.38 | 0.60 | 0.04 | 0.11 | 0.12 | - | - | - |
| | P04 | 1.4819 | 0.19 | 0.43 | 0.63 | 0.01 | 0.11 | 0.25 | - | - | - |
| | P05 | 0.6679 | 0.18 | 0.32 | 0.42 | 0.04 | 0.21 | 0.23 | - | - | - |
| | P06 | 0.7084 | 0.23 | 0.52 | 0.63 | 0.03 | 0.15 | 0.26 | - | - | - |
| | P07 | 0.1862 | 0.08 | 0.32 | 0.36 | 0.56 | 0.64 | 0.64 | - | - | - |
| | P08 | 1.0252 | 0.13 | 0.35 | 0.45 | 0.13 | 0.44 | 0.57 | - | - | - |
| Ours | P01 | 0.4672 | 0.81 | 0.95 | 0.97 | 0.62 | 0.72 | 0.83 | 0.73 | 0.79 | 0.79 |
| | P02 | 0.6462 | 0.65 | 0.73 | 0.75 | 0.63 | 0.75 | 0.77 | 0.62 | 0.77 | 0.79 |
| | P03 | 0.4234 | 0.57 | 0.67 | 0.72 | 0.41 | 0.57 | 0.61 | 0.3 | 0.48 | 0.52 |
| | P04 | 0.3147 | 0.70 | 0.84 | 0.85 | 0.66 | 0.71 | 0.79 | 0.64 | 0.71 | 0.74 |
| | P05 | 0.0117 | 0.59 | 0.80 | 0.82 | 0.76 | 0.86 | 0.88 | 0.76 | 0.83 | 0.90 |
| | P06 | 0.0241 | 0.42 | 0.63 | 0.65 | 0.27 | 0.55 | 0.60 | 0.32 | 0.5 | 0.5 |
| | P07 | 0.1619 | 0.44 | 0.60 | 0.64 | 0.58 | 0.79 | 0.89 | 0.84 | 0.96 | 0.96 |
| | P08 | 0.0199 | 0.47 | 0.65 | 0.68 | 0.39 | 0.82 | 0.87 | 0.82 | 0.89 | 0.89 |

participants, the top-5 accuracy is around or over 80%. Therefore, we include five intentions in the long-term intention list. If the agent finds the user's short-term intention is not in the list, it raises a query to remind the user if anything has been missed.

## 5.5 Discussion

**System Application** The agent predicts the next action sequence. With the prediction of action, the agent further predicts the short-term intention of the user. It also predicts a list of long-term intentions, typically containing five intentions at a time. Then it compares the similarity of short-term and long-term intentions, considering the duration of actions. If the short-term intention is not a son of the long-term intention list, and the duration of the predicted action is too long for the user to address the long-term intention in-time, the agent offers help to the user by interactions. Interactions could be a simple reminder, or some physical assistance with embodied robot.

**Comparation with GPT** We launched a qualitative cooperation between our method and LLMs, namely ChatGPT. We found that ChatGPT is good at inferring short-term intentions but doesn't care much about long-term intentions.

**Future work.** We test the ability of time conflicts. There could be other cases such as inappropriate order of actions (e.g.: putting a cloth into the washing machine before going out). We will discuss these cases in future work.

## 6 Conclusions

In conclusion, our work underscores the importance of imbuing robots with intelligent behavior, particularly in their ability to align with human values. In response to this challenge, we introduced a novel task called "value soft alignment", which involves proactively identifying humans' immediate purpose as well as long-term intentions. To support this task, we constructed a dataset, a cognitive model and an implemented algorithm.

In summary, this work advances the field of human-robot symbiosis by addressing a novel challenge for enhancing robot intelligence, making agents able to cooperate with humans even in situations where human makes mistakes. This work opens up new avenues for creating more intelligent and helpful robotic systems in various applications, ranging from daily companionship to healthcare and beyond.

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

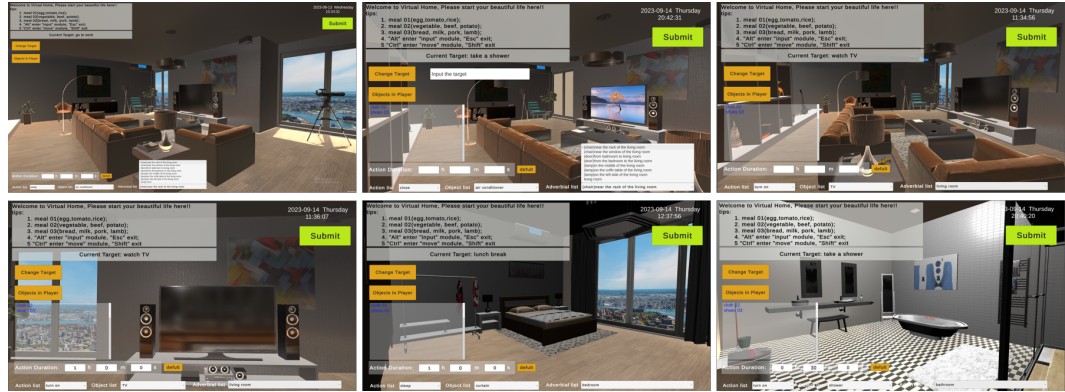

Figure 5: Participants are asked to record their daily behaviors in this virtual house through the user interface. Here are some examples.

Luyao Yuan, Xiaofeng Gao, Zilong Zheng, Mark Edmonds, Ying Nian Wu, Federico Rossano, Hongjing Lu, Yixin Zhu, and Song-Chun Zhu. In situ bidirectional human-robot value alignment. *Science robotics*, 7(68):eabm4183, 2022.

# A APPENDIX

## A.1 SIMULATOR FOR BEHAVIOR DATA COLLECTION

We develop a simulator to collect real human behaviors in the virtual scene. The virtual environment and user interfaces are shown in Figure 5.

## A.2 TRAINING DETAILS

**Hyper-parameters:** Here lists the hyper-parameters used in building the shallow Transformer model. The dimension of the hidden layer in the encoder is 512, which is the same as the decoder. The window size of the encoder input is set to be 14. The learning rate is 1e-05 while the batch size is 32. Learning decay is involved at 10% and batch normalization is added before the input layer, the encoder layer and the decoder layer.

**Data normalization:** Duration data range from 0.5 minutes to over 900 minutes. Using the raw data causes a severe drop in learning performance. We normalize the duration data to make it follow normal distribution $\mathcal{N}(0, 1)$.

## A.3 DATASET DETAILS

The behavior data in the two datasets are formulated in JSON format. The areas of daily activities are limited to six rooms and the interaction furniture of activities is limited to a furniture dictionary, in which the key is the furniture name, and the value is the furniture state class. All of the interaction furniture should be chosen from this dictionary. Actions are also limited to an action dictionary, in which the key is the furniture state class, and the value is the possible action. All actions should be chosen from the value list of the corresponding key.

**Simulated behavior data:** When generating data using GPT, we utilized a shortened description of the behavior. GPT only returns the changed states of objects because of the prompt-length limitation.

Listing 1: Sample of simulated human behavior

```
{
    "target": "get_up",
    "date": "2023.07.24",
    "start_time": "08:20:00",
```

```
 6            "duration": "2",
 7            "action": "wake_up",
 8            "object": [],
 9            "adverbial": "",
10            "player_position": [
11                "bedroom"
12            ],
13            "object_state_change": {}
14        },
15        ...
16        {
17            "target": "morning_wash",
18            "date": "2023.07.24",
19            "start_time": "08:25:00",
20            "duration": "4",
21            "action": "pick_up",
22            "object": [
23                "toothbrush"
24            ],
25            "adverbial": "",
26            "player_position": [
27                "bathroom"
28            ],
29            "object_state_change": {
30                "toothbrush": "with_human"
31            }
32        },
33        {
34            "target": "morning_wash",
35            "date": "2023.07.24",
36            "start_time": "08:29:00",
37            "duration": "1",
38            "action": "put_down",
39            "object": [
40                "toothbrush"
41            ],
42            "adverbial": "",
43            "player_position": [
44                "bathroom"
45            ],
46            "object_state_change": {
47                "toothbrush": "placed"
48            }
49        },
50        ...
51        {
52            "target": "have_breakfast",
53            "date": "2023.07.24",
54            "start_time": "08:37:00",
55            "duration": "15",
56            "action": "eat",
57            "object": [
58                "cooked_meal"
59            ],
60            "adverbial": "on_kitchen_counter",
61            "player_position": [
62                "kitchen"
63            ],
64            "object_state_change": {
65                "cooked_meal": "{\"in_fridge\": 0, \"on_kitchen_shelf\": 0,
                    \"in_sink\": 0, \"on_kitchen_counter\": 0, \"on_dining_
                    table\": 0, \"in_microwave\": 0, \"in_oven\": 0}"
66            }
67        },
68        ...
```

```
69      {
70          "target": "go_to_work",
71          "date": "2023.07.24",
72          "start_time": "08:54:00",
73          "duration": "575.0",
74          "action": "leave_home",
75          "object": [],
76          "adverbial": "",
77          "player_position": [
78              "out_of_home"
79          ],
80          "object_state_change": {}
81      }
```

**Collected behavior data:** We record the full states of all the 206 objects in the collected data. Data in this dataset is far too long to be shown in the paper. To make it clearer, we fold the information of object states in Listing 2.

Listing 2: Sample of collected human behavior

```
1
2      {
3          "target": "watching_tv",
4          "date": "2023.08.29",
5          "start_time": "21:13:09",
6          "duration": "1",
7          "action": "turn_on",
8          "object": "lamp_02",
9          "adverbial": "(lamp)in_the_middle_of_the_living_room",
10         "object_state": {...},
11         "player_position": "living_room"
12     },
13     {
14         "target": "watching_tv",
15             "date": "2023.08.29",
16             "start_time": "21:14:09",
17             "duration": "121",
18             "action": "turn_on",
19             "object": "TV",
20             "adverbial": "living_room",
21             "object_state": {...},
22         "player_position": "living_room"
23     }
24     ...
```

