# OpenReview forum: "Cognitive Modeling for Human-Robot Value Soft Alignment"
_ICLR.cc/2024/Conference — ICLR 2024 Conference Withdrawn Submission_

### Official Review · Reviewer_GZXC · 2023-10-25

**Soundness:** 1 poor
**Presentation:** 1 poor
**Contribution:** 2 fair
**Rating:** 1
**Confidence:** 4

**Summary:**

This research addresses the need for robots to understand human behavior, even when humans make mistakes. The paper introduces a task called "value soft alignment," where robots proactively identify potentially hazardous human actions and offer appropriate suggestions. To accomplish this, the authors propose a value-driven cognition model and a two-stage method that predicts long-term intentions and compares them to short-term actions.

**Strengths:**

- The authors propose an important problem that has far reaching implications across research in designing personalized virtual agents and human-robot collaboration.

- Their proposed dataset could be potentially useful in modeling long term human activities.

**Weaknesses:**

- Unclear Terminology: According to this paper, ``An intelligent robot is expected to align with the values reflected by these long-term behavioral patterns, rather than directly assisting humans with immediate tasks that may contradict long-term human values. Such a robot demonstrates the ability to softly align with human values, which we denote as value soft alignment.`` However, in their method, do not explicitly model long term behavior patterns in conjunction with short term behaviors and deal with conflicting scenarios. Therefore, it is hard to judge if this paper is more of a theoretical position paper or a paper that contributes a novel method or dataset. This paper also introduces terms like hard alignment and soft alignment, but does not formalize them or propose a good evaluation to judge if any such alignment occurs between a human and an AI agent. In a similar vein, the term `soft value alignment` seems to be used loosely in many places. This makes it very tricky to judge this paper’s contributions as they claim to introduce this new task, and not having a formal definition of that can be very misleading.

- Lack of a Baseline: The paper says: `` Since value soft alignment is a cognitive process, we refer to cognitive architectures when modeling human value. There has been a variety of cognitive architectures such as Belief-Desire-Intention (BDI) (Harnad, 1990), Learning Intelligent Distribution Agent (LIDA) (Faghihi & Franklin, 2012), Simulation of the Mental Apparatus & Applications (SiMA) (Schaat et al., 2015), Adaptive Control of Thought-Rational (ACT-R) (Ritter et al., 2019), etc. AGI models demonstrate high-level cognitive intelligence when incorporating values (Peng et al., 2023). Consequently, a value-based cognitive model holds significant potential for managing complex tasks. Following these, we propose our value-based cognitive model.`` If they claim that value soft alignment is a cognitive process, they do not compare it to these other cognitive architectures, nor do they compare their method with any other baselines.

**Questions:**

- Since this paper was submitted to a machine learning conference, the authors should consider focusing on writing more details about the method, justifying what worked and why, a strong evaluation (with more subjects, against more baselines and evaluation metrics) over explaining repeatedly the necessity of why this work needs to be done.

- The authors should reconsider the claims they make in the paper, as all they have done is collected some data on a simulator with humans and ChatGPT and run an off the shelf multi-head transformer model to model the sequence of actions taken by a human. However, they claim that `We build a value-driven cognitive model to tackle the proposed “value soft alignment” tasks, which allows the robot to learn long-term behavioral patterns guided by human values and provide immediate suggestions for potentially unreasonable short-term behaviors.`  There seems to be a disconnect between what work has been done and what has been claimed.

- The authors should consider adding anecdotal examples from their experiments with people to help the readers understand intuitively how their method works. This should have details about the task, what the human did, how their model responded, and how alignment between the human and the agent was achieved.

- The authors should expand about what other datasets are similar to theirs, and why theirs is better. Eg: Ego4D

- Is Figure 1 the output of their method, or just a running example. According to Figure 2, the dataset contains data of multiple days and it is unclear if they actually collected data for multiple days or if it is a proposal. The inputs and outputs in Figure 3 are unclear.

---

### Official Review · Reviewer_zjY7 · 2023-10-26

**Soundness:** 1 poor
**Presentation:** 2 fair
**Contribution:** 1 poor
**Rating:** 3
**Confidence:** 5

**Summary:**

This paper introduced a newly collected dataset for short-term and longer-term intention recognition and value alignment. The paper also implements a cognitive model to learn long-term behavior patterns.

**Strengths:**

+ The problem of value alignment can be an important topic for human-robot collaboration.
+ Open sourcing the collected dataset can be beneficial.

**Weaknesses:**

- The paper has a low theoretical novelty. The proposed value-based cognitive model argues to model long-term intention simply using an extended time window. It does not provide any new ideas.

- The motivation example on the cooking and later attending an appointment is not convincing to motivate this work or show challenges. It sounds like a problem that can be easily addressed by a reminder app.

- The paper argues to align an agent’s value with a human. But it seems the proposed model only focuses on recognizing human behaviors.

- The paper argues to enable a robot’s capability. However, the approach and dataset are not relevant to robots.

**Questions:**

See the weakness section.

---

### Official Review · Reviewer_rkNb · 2023-11-02

**Soundness:** 3 good
**Presentation:** 3 good
**Contribution:** 2 fair
**Rating:** 5
**Confidence:** 4

**Summary:**

This work proposes the problem of human value soft alignment, which is to align with long-term high-level human goals rather than their immediate actions and short-term plans. The work proposes a method to model both high-level human intention and short-term actions, and evaluate the method with synthetic data against baselines.

**Strengths:**

1. Problem is well-motivated: It is grounded in the extensive study of human intention understanding, and the problem of human value soft alignment is important and interesting.

2. The method and architecture designs are clear and well presented.

3. The evaluation metric is well-defined, and experiments are conducted with rigor.

**Weaknesses:**

1. The work is motivated by the notion that human behaviors can be suboptimal and can conflict with their higher level intent. However, there are no real examples to prove this point. It is still unclear if this problem exists in the actual experiment conducted, and if there is correlation between the high-level and low-level targets.

2. Data: The data is mainly generated by GPT, and only a small portion is human data. It is dubious how GPT generated data reflect the true distribution of human behaviors in the real world. GPT models are known to have hallucinations and can display unreasonable behavior, which limits the real-world application of the work.

3. A related point on data: for the simulated data by GPT, they are not real interaction data of an embodied agent in a real interactive environment. In that case, how is the duration of interaction determined then? There seems to be no real “ground truth” standard and it is hard to see how the distribution of the synthesis data matches that of the real human beings.

**Questions:**

1. Could you provide some examples of traces of human action data, and some concrete examples of how high-level targets can conflict with low-level targets? I get the motivation but I’d like to see how concrete this problem is in realistic settings.

2. Why is predicting action duration important and used as a metric? It looks like value alignment concerns more about goals and human intentions, rather than how exactly a human does it.

3. How is the duration of an action modeled (how is it known what is the actual duration of a task?) - does it depend on some intuitive standard like how completed a task has been performed?

4. In Table 1: What is there no long-term target prediction results for the end-to-end approach?

5. Why is the training data generated by GPT3.5 and testing data generated by GPT4? The output of these two models are pretty different. How much is the distribution shift of the action behaviors of the two?

6. It is mentioned that GPT generates ghost data - where is the example in the Appendix?